# Socioeconomic inequalities in high-risk fertility behaviour among Nigerian women: A non-experimental population-based study

**Michael Ekholuenetale**●*

Faculty of Science and Health, School of Health and Care Professions, University of Portsmouth, Hampshire, United Kingdom

* mic42006@gmail.com

## Abstract

### Background

High-risk fertility behaviour (HRFB) has adverse health, social and economic effects on women. An understanding of socioeconomic inequalities is needed to design effective interventions targeted to lower maternal morbidity and mortality due to HRFB.

### Objectives

The objective was to quantify socioeconomic inequalities in HRFB among Nigerian women.

### Design

This was a cross-section study that used population-based data.

### Methods

A sample of 21,792 Nigerian women aged 15–49 years from the 2018 Nigeria Demographic Health Survey was analyzed. Percentage was employed in univariate analysis. In addition, concentration index was used to assess the extent of inequalities in HRFB. This was further decomposed to ascertain the explanatory components' relative contributions to the socioeconomic inequalities in HRFB.

### Results

The weighted prevalence of HRFB (63.5%; 95% CI: 62.6–64.4%), included <18 years at childbirth (4.9%; 95% CI:4.5–5.3%), >34 years at childbirth (18.3%; 95% CI: 17.6–19.0%), >3 children birth order (49.6%; 95% CI: 48.7–50.5%) and <24 months at preceding birth interval (17.0% 95% CI: 16.3–17.6%) were estimated. Education (Contri: 50.3997%, $E_c$: 0.2771), wealth (Contri: 27.2811%, $E_c$: 0.2665), socioeconomic disadvantaged (Contri: 14.9163%, $E_c$: -0.0996), religion (Contri: 13.8636%, $E_c$: -0.0496), region (Contri: 11.1724%, $E_c$: 0.0711), partner education (Contri: 7.1351%, $E_c$: 0.2138), media use (Contri: 4.5064%, $E_c$: 0.1449) and family motility (Contri: 3.7890%, $E_c$: -0.0281) were positive contributors to HRFBs among Nigerian women. However, age (Contri: -14.6237, $E_c$: 0.0089) and history of

**Data Availability Statement:** Data for this study were sourced and available here: https://dhsprogram.com/data/dataset/Nigeria_Standard-DHS_2018.cfm?flag=1. Data may be requested

from The DHS Program after creating an account and submitting a concept note. More access information can be found on The DHS Program website (https://dhsprogram.com/data/Access-Instructions.cfm). The authors confirm that interested researchers would be able to access these data in the same manner as the authors. The authors also confirm that they had no special access privileges that others would not have.

**Funding:** The author(s) received no specific funding for this work.

**Competing interests:** The authors have declared that no competing interests exist.

contraceptive use (Contri: -8.7723, $E_c$: -0.2094) were negative contributors to HRFBs among women of reproductive age in Nigeria.

## Conclusion

Women who have low socioeconomic level reported higher prevalence of HRFB. Targeted interventions are required to lower HRFB among Nigerian women from poor households and with no formal education. Women should get counselling and assistance from health-care and educational institutions to help them adopt healthy sexual and reproductive practices.

## Introduction

Every year, pregnancy-related complications account for more than half of all women's deaths in resource-constrained settings, where over 830 women die each day from pregnancy or childbirth-related issues worldwide [1]. According to the World Health Organization (WHO), 9 out of every 10 maternal deaths occur in developing countries [2]. In 2015, about 303,000 maternal deaths were reported globally [1]. The majority of these deaths were reported in low-resource countries, which could have been prevented, if adequate sexual and reproductive health services and information were provided or utilized as per international recommendations [1, 3–7]. This especially could continue to be worsen in low-resource settings, as disadvantaged women have been reported to have staggering estimates of morbidity and mortality, as well as low uptake of maternal health services [8–11]. Additionally, when these health disparities are unjust, unnecessary, and avoidable, they result in health inequity among the populace [12].

Evidence suggests that disadvantaged women, such as the majority from resource-constrained settings have less access to sexual and reproductive health care information and services, which could be responsible for poor maternal health indices including high morbidity and mortality rates [13–15]. Inequalities and inequities in health services utilization have been reported in several studies conducted in low-resource settings [16–19]. Women's sexual and reproductive health require significant attention in global scheme of health interventions [4]. The first target of the third Sustainable Development Goal (SDG 3.1) aims to reduce maternal mortality ratio to less than 70 per 100 000 live births by 2030 [20]. Additionally, the third and tenth SDGs are targeted to reduce inequalities: "ensure healthy lives and promote well-being for all at all ages" and "reduce inequality within and among countries" [20]. To achieve these, evidence-based studies are required to determine the extent of coverage and inequalities in sexual and reproductive health indicators particularly in resource-constrained settings.

High-risk fertility behaviour (HRFB) is a risk factor for maternal mortality [21]. It poses significant risks to maternal and child health. These behaviours exist and include early or late childbearing, high parity and short interbirth interval. These indicators significantly increase the risks of adverse maternal and child health outcomes, including maternal mortality, infant mortality, preterm birth and low birth weight amongst others [22]. In Nigeria, the persistence of HRFB is a critical public health concern, exacerbated by socioeconomic inequalities. Socio-economic inequalities play a pivotal role in shaping HRFB, as they influence access to health-care services and reproductive health outcomes. Studies have consistently shown that women with higher educational attainment are more likely to delay childbearing, have fewer children and space their births more appropriately [23]. Women with no formal education are

significantly more likely to engage in early childbirth, high parity and short interbirth interval [24]. Education empowers women with knowledge about reproductive health, access to family planning services and the ability to make informed decisions about their fertility.

The wealth index, which combines various indicators of economic status, is a strong predictor of HRFB. Studies indicate that women from wealthier households tend to have fewer children and longer birth intervals compared to those from poorer households [25]. Wealthier women have greater access to healthcare services, including prenatal and postnatal care, contraception, and education, which collectively reduce the likelihood of engaging in HRFB [26]. However, patriarchal norms and gender roles often limit women's autonomy in making reproductive decisions [27]. In many Nigerian communities, early marriage and high fertility are common, leading to higher rates of HRFB among women. Additionally, societal expectations and pressure to prove fertility can drive women to have many children in quick succession, exacerbating the risks associated with HRFB. The sub-optimal healthcare system [28], also plays a crucial role in mediating the relationship between socioeconomic status and HRFB.

Access to quality maternal healthcare services and information is often unevenly distributed, with poorer and uneducated women facing significant barriers. These barriers include long distances to healthcare facilities, high costs of services, inadequate information and counselling and inadequate healthcare infrastructure. Improving the accessibility and quality of maternal healthcare services is essential in reducing HRFB among socioeconomically disadvantaged women. Education, income, employment, place of residence and wealth index are regarded as critical socioeconomic determinants of HRFB. Addressing these inequalities requires comprehensive interventions that promote education, economic opportunities, healthcare access and community engagement. By tackling the root causes of socioeconomic inequalities, Nigeria can improve reproductive health outcomes and reduce the prevalence of HRFB, ultimately contributing to better maternal and child health. The objective of this study was to examine the socioeconomic inequalities in HRFB among Nigeria women.

## Methods

### Data source

Data from the 2018 Nigeria demographic and health survey (NDHS) individual woman questionnaire was used. In total, 21,792 women who were of reproductive age made up the study's sample. The National Population Commission (NPC) has conducted this type of survey six times, with the most recent being the 2018 NDHS. From August 14 to December 29, 2018, data were collected [29]. The sample was chosen using a stratified, two-stage cluster design, with enumeration areas (EAs) serving as the sampling units for the first stage. A total of approximately 30 households were chosen from the complete list of households in each of the 1,389 selected EAs, resulting in a 99% response rate.

### Sampling technique

In the three-stage sampling stratification process used for the NDHS 2018, respondents were first divided into urban and rural housing strata, and then EAs were randomly chosen within each stratum. Following that, equal probability sampling was used to choose households within each EA for the survey. In order to ensure that the sample was representative of the general population, the three-stage sampling method was used when calculating survey weights. The Federal Republic of Nigeria's 2006 Population and Housing Census (NPHC), which was carried out by the National Population Commission, served as the sampling frame for the 2018 NDHS. A stratified sample was chosen in two phases for the 2018 NDHS. The 36 states and the Federal Capital Territory were divided into urban and rural regions in order to stratify the country. There were 74 different

sampling strata identified in total. The individual female data used for analysis in this study served as the source of the data. Since its inception in 1984, the DHS project, which is primarily funded by the United States Agency for International Development (USAID) with assistance from other donors and host countries, has carried out over 230 nationally representative and globally comparable household surveys in more than 80 countries. The data can be accessed at: https://dhsprogram.com/data/dataset/Nigeria_Standard-DHS_2018.cfm?flag=1. Information regarding the DHS sampling process has previously been reported [30].

## Measurements of outcome variable

HRFB was the outcome variable and computed based on four parameters: (I) women aged <18 years at the time of childbirth; (II) women aged >34 years at the time of childbirth; (III) women of high parity (>3 children) and; (IV) women of a child born after a short birth interval (<24 months); and If a woman experienced at least one of the indicators, she was defined as having experienced HRFB, coded as 1 or "yes," otherwise 0 or "no" [25, 31, 32].

## Social determinants of health (SDoH) framework

The social determinants of health (SDoH) framework highlights how broader social and economic factors influence health outcomes. The social and economic opportunities as well as the resources and supports available in women's homes, neighbourhoods, and communities are included in SDoH [33]. This framework was the basis for selecting the risk factors of HRFB to understand the interactions between various factors that contribute to HRFB. The key risk factors for HRFB include poverty, low income, and economic dependency which could lead to higher fertility due to limited access to healthcare. In addition, low educational attainment is often linked to early childbearing and lack of family planning use. Evidence suggests that these factors influence women's health by shaping barriers and facilitators to health-related behaviours [34]. Based on SDoH framework, socio-economic, demographic, healthcare, and physical settlement variables were selected in this study.

## Explanatory variables

Family motility: < 5 years vs. 5+ years (native); Age (years): 15-19/20-24/25-29/30-34/35-39/40-44/45-49; Region: North Central/North East/North West/South East/South South/South West; Residence: Urban vs. Rural; Education: No education/Primary/Secondary/Higher; Religion: Christian/Islam/Traditional/others; Marital status: Single/Married; Ever used contraceptive method: No vs. Yes; Covered by health insurance: No vs. Yes; Media use: No vs. Yes; Employment status: No vs. Yes; Sex of household head: Male vs. Female; Socioeconomic disadvantaged status: Low/Medium/High; Partner education: No education/Primary/Secondary/Higher/Do not know. Household wealth: Poorest/Poorer/Middle/Richer/Richest. In creating household wealth index, DHS adjusted for rural-urban differences which was used for analysis. As a response to criticism that a single wealth index is too urban in its construction and not able to distinguish the poorest of the poor from other poor households, the new variable created to provide an urban- and rural-specific wealth index was utilized.

## Analytical approach

Data analysis was conducted using Stata software version 14.0 (Stata Corporation, College Station, Texas, USA). Utilising the survey module's ('svy') command to account for sampling design, the analysis in Stata took into account the multi-stage stratified cluster sample design. In the univariate analysis, percentage was used. In HRFB, concentration indices and Lorenz

curves were used to investigate wealth-, and education-based inequalities. When HRFB was higher among wealthy or educated women, the concentration index value was positive. Conversely, a negative concentration index value indicates otherwise [35, 36]. The level of statistical significance was set at $p < 0.05$.

## Concentration curves and indices

The analysis of health inequalities frequently employs the concentration index. The existence of health inequalities is examined by the indices and curves. The Erreygers normalised concentration indices [37] were used in this study to assess the degree of education-based inequalities in HRFB. The Erreygers was chosen over the other possible indices because of its decompossability and simplicity. The 'convenient covariance' can be used to compute the concentration index, as shown below:

$$CI = \frac{2}{\hat{y}} COV(y_i, R_i) \tag{1}$$

Where: $y_i$ is the health variable

$\hat{y}$ is the mean of $y_i$

$R_i$ is the fractional rank of the ith individual

COV symbolizes the covariance

By dividing by two the distance between the concentration curve and the line of equality (the 45-degree line), concentration indices are calculated [38]. If the concentration curve is on the 45˚ line, there is no health disparity. The magnitude of the health disparity is indicated by the concentration curve's angle from the line of equality (45˚ line). The degree of health inequality increases with the width of the gap between the concentration curve and the line of equality. This study chose to use normalised formulae because it has been suggested that doing so ensures that the boundaries problem for a binary Cardinal Health variable is resolved. The Erreygers normalized index (E(c)) is denoted as:

$$E_c = \frac{4\hat{y}}{y^{max} - y^{min}} CI \tag{2}$$

In the case of binary variables, $y^{max} - y^{min}$ represents the range of the health variable, which is 'one'. The current study focused on the Erreygers normalised index because it is the corrected concentration index that is most commonly used in the health literature.

## Decomposing Erreygers Normalised Concentration Index

To determine the contributions of women's health indicator determinants, the Erreygers Normalised Concentration Index can be decomposed [39, 40]. Each explanatory factor's contribution to health inequalities was broken down into its component parts by health elasticity. Given a linear relationship between individual health (yi) and a collection of k explanatory variables, yi will be as follows:

$$y_i = a + \sum_k \beta_k X_{ki} + \varepsilon_i \tag{3}$$

Wagstaff et al. [40] show the concentration index for any health measure that has a linear relationship with a set of k exploratory variables may be divided as follows:

$$CI = \sum_k \left( \frac{\beta_k \dot{x}_k}{\hat{y}} \right) CI_k + \frac{GCI_\varepsilon}{\hat{y}} \tag{4}$$

Where: $\beta_k$ is the partial

$\hat{y}$ is the mean of the health variable

$\dot{x}_k$ is the mean of $\dot{x}_k$

$CI_k$ denotes the concentration index of $x_k$ against education

$GC_\varepsilon$ is the generalised concentration for the error term

To decompose the Erreygers concentration index, we modified Eq (4) as shown below [41]:

$$E_c = 4[\sum_k (\beta_k \dot{x}_k)CI_k + GCI_\varepsilon] \tag{5}$$

## Ethical consideration

For this study, the secondary dataset that was available to the general public had identifiers removed. Following a recognized ethical procedure, NDHS obtained the respondents' informed consent. No additional participants' consent was needed because the authors were given permission to use this data which was collected following ethical standards. Here is where you can find details on DHS guidelines: http://goo.gl/ny8T6X.

## Results

Table 1 showed the weighted prevalence of HRFBs (63.5%; 95% CI: 62.6–64.4%), which included <18 years at childbirth (4.9%; 95% CI:4.5–5.3%), >34 years at childbirth (18.3%; 95% CI: 17.6–19.0%), >3 children birth order (49.6%; 95% CI: 48.7–50.5%) and <24 months at preceding birth interval (17.0% 95% CI: 16.3–17.6%). The weighted prevalence of HRFBs was higher among women in rural residence, with no formal education, those from poorest households and highly socioeconomically disadvantaged. The prevalence distribution of HRFBs varied across respondents' characteristics. See Table 1 for the details.

Fig 1 showed the wealth-based inequalities in HRFBs, <18 years at childbirth, >34 years at childbirth, 4+ birth order and <24 months preceding birth interval among women of reproductive age in Nigeria. How far the curves deviated from the line of equality (diagonal line) indicated whether there were inequalities in HRFBs, <18 years at childbirth, >34 years at childbirth, 4+ birth order and <24 months preceding birth interval and to what extent. This showed that poor women had higher HRFBs, <18 years at childbirth, >34 years at childbirth, 4+ birth order and <24 months preceding birth interval as the line of equality sagged above the diagonal line.

Fig 2 showed the education-based inequalities in HRFBs, <18 years at childbirth, >34 years at childbirth, 4+ birth order and <24 months preceding birth interval among women of reproductive age in Nigeria. How far the curves deviated from the line of equality (diagonal line) indicated whether there were inequalities in HRFBs, <18 years at childbirth, >34 years at childbirth, 4+ birth order and <24 months preceding birth interval and to what extent. This showed that women with no formal education had higher HRFBs, <18 years at childbirth, >34 years at childbirth, 4+ birth order and <24 months preceding birth interval as the line of equality sagged above the diagonal line.

Table 2 showed results of wealth and education-based inequalities in HRFBs Nigerian among women. Across the levels of women's characteristics, the results showed higher HRFBs among poor women and those with no formal education. Moreover, there was a difference in the concentration indices across the levels of the following variables: age, region, residence, religion, marital status, health insurance coverage, media use, employment, sex of household head, neighbourhood socioeconomic disadvantaged and partner education respectively. See Table 2 for the details.

**Table 1. Distribution of HRFBs by women's characteristics (n = 21,792).**

| Variable | n (%) | HRFBs, % (95% CI) | <18 years at childbirth, % (95% CI) | >34 years at childbirth, % (95% CI) | >3 children birth order, % (95% CI) | <24 months at preceding birth interval, % (95% CI) |
|---|---|---|---|---|---|---|
| **Family motility** | | | | | | |
| < 5 years | 3578 (16.4) | 43.8 (41.8–45.8) | 7.4 (6.4–8.5) | 10.1 (8.9–11.4) | 22.2 (20.5–24.0) | 16.2 (14.8–17.8) |
| 5+ years (native) | 18241 (83.6) | 67.4 (66.5–68.3) | 4.4 (4.0–4.8) | 19.9 (19.2–20.7) | 55.0 (54.1–56.0) | 17.1 (16.4–17.8) |
| **Age (years)** | | | | | | |
| 15–19 | 1193 (5.5) | 71.8 (68.5–74.8) | 68.3 (65.0–71.5) | 0.0 | 0.0 | 6.6 (5.2–8.3) |
| 20–24 | 4206 (19.3) | 27.9 (25.8–30.1) | 5.7 (4.8–6.7) | 0.0 | 8.3 (7.4–9.4) | 17.6 (16.1–19.3) |
| 25–29 | 5617 (25.8) | 50.3 (48.6–51.9) | 0.0 | 0.0 | 40.4 (38.7–42.1) | 19.8 (18.5–21.1) |
| 30–34 | 4670 (21.4) | 70.6 (68.8–72.4) | 0.0 | 0.0 | 65.6 (63.6–67.6) | 18.3 (17.0–19.7) |
| 35–39 | 3622 (16.6) | 89.3 (87.9–90.6) | 0.0 | 44.1 (42.1–46.1) | 79.0 (77.1–80.8) | 16.2 (14.8–17.7) |
| 40–44 | 1774 (8.1) | 100.0 | 0.0 | 100.0 | 91.6 (89.8–93.2) | 13.4 (11.6–15.4) |
| 45–49 | 710 (3.3) | 100.0 | 0.0 | 100.0 | 94.5 (92.0–96.2) | 12.6 (10.2–15.4) |
| **Region** | | | | | | |
| North Central | 3875 (17.8) | 58.4 (56.2–60.6) | 3.8 (3.1–4.6) | 15.6 (14.3–17.1) | 45.9 (43.6–48.1) | 15.1 (13.6–16.7) |
| North East | 4506 (20.7) | 69.5 (67.6–71.3) | 6.9 (5.9–8.0) | 16.8 (15.5–18.2) | 55.7 (53.8–57.6) | 18.9 (17.4–20.6) |
| North West | 6309 (29.0) | 72.2 (70.7–73.7) | 6.9 (6.1–7.7) | 18.8 (17.4–20.2) | 59.6 (58.0–61.2) | 17.7 (16.5–19.0) |
| South East | 2365 (10.9) | 62.2 (60.2–64.1) | 2.2 (1.7–2.9) | 20.4 (18.7–22.3) | 44.7 (42.0–47.3) | 21.8 (20.1–23.7) |
| South South | 2174 (10.0) | 55.2 (52.5–57.7) | 3.3 (2.4–4.5) | 18.4 (16.4–20.5) | 38.1 (35.6–40.7) | 17.7 (15.9–19.6) |
| South West | 2563 (11.8) | 46.7 (44.4–48.9) | 1.5 (1.1–2.1) | 20.1 (18.4–21.8) | 32.5 (30.1–35.0) | 11.1 (9.9–12.5) |
| **Residence** | | | | | | |
| Urban | 7710 (35.4) | 58.2 (56.7–59.6) | 2.4 (1.9–2.9) | 19.8 (18.7–20.9) | 43.7 (42.2–45.3) | 16.5 (15.5–17.4) |
| Rural | 14082 (64.6) | 67.0 (66.0–68.1) | 6.5 (6.0–7.1) | 17.3 (16.5–18.2) | 53.5 (52.4–54.6) | 17.3 (16.5–18.2) |
| **Education** | | | | | | |
| No education | 9527 (43.7) | 74.6 (73.6–75.6) | 6.8 (6.2–7.5) | 20.6 (19.4–21.7) | 62.5 (61.3–63.7) | 18.2 (17.2–19.2) |
| Primary | 3410 (15.7) | 70.5 (68.4–72.6) | 4.7 (3.7–5.9) | 22.6 (21.0–24.4) | 59.3 (57.2–61.3) | 15.8 (14.3–17.3) |
| Secondary | 7064 (32.4) | 49.6 (48.2–51.0) | 3.5 (3.1–4.1) | 12.8 (11.8–13.8) | 33.9 (32.4–35.2) | 16.0 (14.9–17.1) |
| Higher | 1791 (8.2) | 45.6 (42.8–48.4) | 0.04 (0.01–18.) | 19.5 (17.5–21.8) | 24.6 (22.2–27.2) | 16.8 (14.9–18.9) |
| **Household wealth** | | | | | | |
| Poorest | 4652 (21.4) | 73.1 (71.5–74.6) | 6.9 (6.0–7.9) | 21.2 (19.8–22.7) | 60.5 (58.8–62.2) | 17.8 (16.6–19.1) |
| Poorer | 4613 (21.2) | 67.2 (65.5–69.0) | 5.5 (4.8–6.3) | 17.7 (16.3–19.1) | 54.9 (53.2–56.6) | 17.7 (16.4–19.1) |
| Middle | 4462 (20.5) | 61.9 (59.9–63.9) | 5.3 (4.5–6.1) | 15.6 (14.2–17.0) | 48.4 (46.5–50.3) | 16.2 (14.9–17.6) |

*(Continued)*

**Table 1.** (Continued)

| Variable | n (%) | HRFBs, % (95% CI) | <18 years at childbirth, % (95% CI) | >34 years at childbirth, % (95% CI) | >3 children birth order, % (95% CI) | <24 months at preceding birth interval, % (95% CI) |
|---|---|---|---|---|---|---|
| Richer | 4269 (19.6) | 58.8 (56.8–60.7) | 4.3 (3.4–5.4) | 17.4 (16.0–19.0) | 44.9 (43.0–46.8) | 15.7 (14.4–17.1) |
| Richest | 3796 (17.4) | 55.3 (53.2–57.4) | 1.9 (1.4–2.5) | 20.1 (18.5–21.8) | 37.5 (35.4–39.5) | 17.6 (16.0–19.4) |
| **Religion** | | | | | | |
| Christian | 8929 (41.0) | 54.1 (52.8–55.3) | 2.6 (2.2–3.0) | 18.7 (17.8–19.7) | 38.7 (37.4–40.0) | 16.0 (15.1–17.0) |
| Islam | 12687 (58.2) | 69.3 (68.2–70.4) | 6.3 (5.7–6.9) | 18.0 (17.1–19.0) | 56.3 (55.2–57.5) | 17.6 (16.8–18.5) |
| Traditional/others | 176 (0.8) | 71.0 (60.1–79.9) | 4.6 (2.1–9.5) | 20.6 (14.3–28.6) | 55.5 (42.7–67.5) | 16.3 (11.0–23.5) |
| **Marital status** | | | | | | |
| Single | 605 (2.8) | 29.1 (24.8–33.8) | 16.6 (13.4–20.5) | 4.7 (3.0–7.4) | 6.8 (4.7–9.6) | 5.0 (3.2–7.8) |
| Married | 21187 (97.2) | 64.3 (63.4–65.2) | 4.6 (4.2–5.0) | 18.6 (18.0–19.3) | 50.6 (49.7–51.6) | 17.3 (16.6–17.9) |
| **Ever used contraceptive method** | | | | | | |
| No | 14897 (68.4) | 64.4 (63.4–65.4) | 6.4 (5.9–6.9) | 17.6 (16.8–18.4) | 50.0 (49.0–51.0) | 16.8 (16.1–17.6) |
| Yes | 6895 (31.6) | 61.7 (60.3–63.2) | 1.7 (1.4–2.1) | 19.8 (18.7–21.0) | 48.8 (47.1–50.4) | 17.3 (16.2–18.4) |
| **Covered by health insurance** | | | | | | |
| No | 21298 (97.7) | 63.7 (62.8–64.6) | 4.9 (4.6–5.4) | 18.2 (17.6–18.9) | 49.9 (49.0–50.8) | 17.0 (16.3–17.6) |
| Yes | 494 (2.3) | 57.0 (51.0–62.9) | 1.4 (0.5–4.0) | 21.3 (16.9–26.4) | 37.2 (31.7–43.0) | 18.4 (14.2–23.7) |
| **Media use** | | | | | | |
| No | 8470 (38.9) | 71.0 (69.8–72.2) | 7.5 (6.8–8.3) | 17.4 (16.4–18.5) | 57.3 (56.0–58.6) | 18.2 (17.2–19.3) |
| Yes | 13322 (61.1) | 59.0 (58.0–60.1) | 3.3 (2.9–3.7) | 18.9 (18.1–19.7) | 45.0 (43.9–46.2) | 16.3 (15.5–17.0) |
| **Employment status** | | | | | | |
| No | 6977 (32.0) | 61.2 (59.7–62.6) | 8.9 (8.1–9.7) | 12.2 (11.3–13.2) | 43.9 (42.5–45.3) | 18.1 (16.9–19.3) |
| Yes | 14815 (68.0) | 64.6 (63.5–65.7) | 3.0 (2.7–3.4) | 21.1 (20.3–22.0) | 52.3 (51.1–53.4) | 16.5 (15.8–17.2) |
| **Sex of household head** | | | | | | |
| Male | 19512 (89.5) | 64.5 (63.5–65.4) | 4.9 (4.5–5.3) | 18.2 (17.5–18.9) | 50.7 (49.7–51.6) | 17.3 (16.7–18.1) |
| Female | 2280 (10.5) | 54.5 (52.0–57.0) | 4.6 (3.7–5.7) | 19.6 (17.8–21.6) | 40.0 (37.8–42.3) | 13.7 (12.1–15.4) |
| **Socioeconomic disadvantaged status** | | | | | | |
| Low | 6509 (29.9) | 53.3 (51.7–54.9) | 1.7 (1.3–2.1) | 19.5 (18.4–20.7) | 37.8 (36.0–39.6) | 15.7 (14.7–16.7) |
| Medium | 6058 (27.8) | 62.6 (60.7–64.5) | 3.6 (3.0–4.2) | 18.6 (17.4–19.8) | 50.5 (48.8–52.3) | 16.5 (15.4–17.6) |
| High | 9225 (42.3) | 71.7 (70.6–72.8) | 8.0 (7.3–8.7) | 17.2 (16.2–18.4) | 58.0 (56.6–59.3) | 18.2 (17.1–19.4) |
| **Partner education** | | | | | | |

*(Continued)*

**Table 1.** (Continued)

| Variable | n (%) | HRFBs, % (95% CI) | <18 years at childbirth, % (95% CI) | >34 years at childbirth, % (95% CI) | >3 children birth order, % (95% CI) | <24 months at preceding birth interval, % (95% CI) |
|---|---|---|---|---|---|---|
| No education | 7141 (35.0) | 74.9 (73.7–76.0) | 6.9 (6.3–7.7) | 20.0 (18.7–21.4) | 62.4 (61.0–63.8) | 18.6 (17.4–19.8) |
| Primary | 2897 (14.2) | 71.3 (69.3–73.2) | 4.7 (3.7–5.9) | 22.0 (20.2–23.9) | 58.7 (56.6–60.8) | 17.8 (16.2–19.4) |
| Secondary | 7060 (34.6) | 56.3 (54.7–57.6) | 3.2 (2.7–3.8) | 16.0 (14.9–17.1) | 42.0 (40.4–43.7) | 16.6 (15.7–17.6) |
| Higher | 3039 (14.9) | 51.7 (49.2–54.1) | 1.5 (1.1–2.1) | 17.8 (16.0–19.7) | 35.2 (32.7–37.8) | 16.0 (14.3–17.7) |
| Do not know | 282 (1.4) | 68.6 (61.5–74.8) | 6.2 (3.8–10.1) | 16.3 (12.0–21.7) | 57.3 (51.8–62.6) | 17.9 (11.9–26.0) |
| **Total estimate** | 21792 (100.0) | 63.5 (62.6–64.4) | 4.9 (4.5–5.3) | 18.3 (17.6–19.0) | 49.6 (48.7–50.5) | 17.0 (16.3–17.6) |

Table 3 showed Erreygers concentration (Ec) indices decomposed in order to determine the contribution (Contri) of HRFBs among women of reproductive age in Nigeria. Education (Contri: 50.3997%, $E_c$: 0.2771), wealth (Contri: 27.2811%, $E_c$: 0.2665), neighbourhood socio-economic disadvantaged (Contri: 14.9163%, $E_c$: -0.0996), religion (Contri: 13.8636%, $E_c$: -0.0496), region (Contri: 11.1724%, $E_c$: 0.0711), partner education (Contri: 7.1351%, $E_c$: 0.2138), media use (Contri: 4.5064%, $E_c$: 0.1449) and family motility (Contri: 3.7890%, $E_c$: -0.0281) were positive contributors to HRFBs among Nigerian women. However, age (Contri: -14.6237, $E_c$: 0.0089) and history of contraceptive use (Contri: -8.7723, $E_c$: -0.2094) were

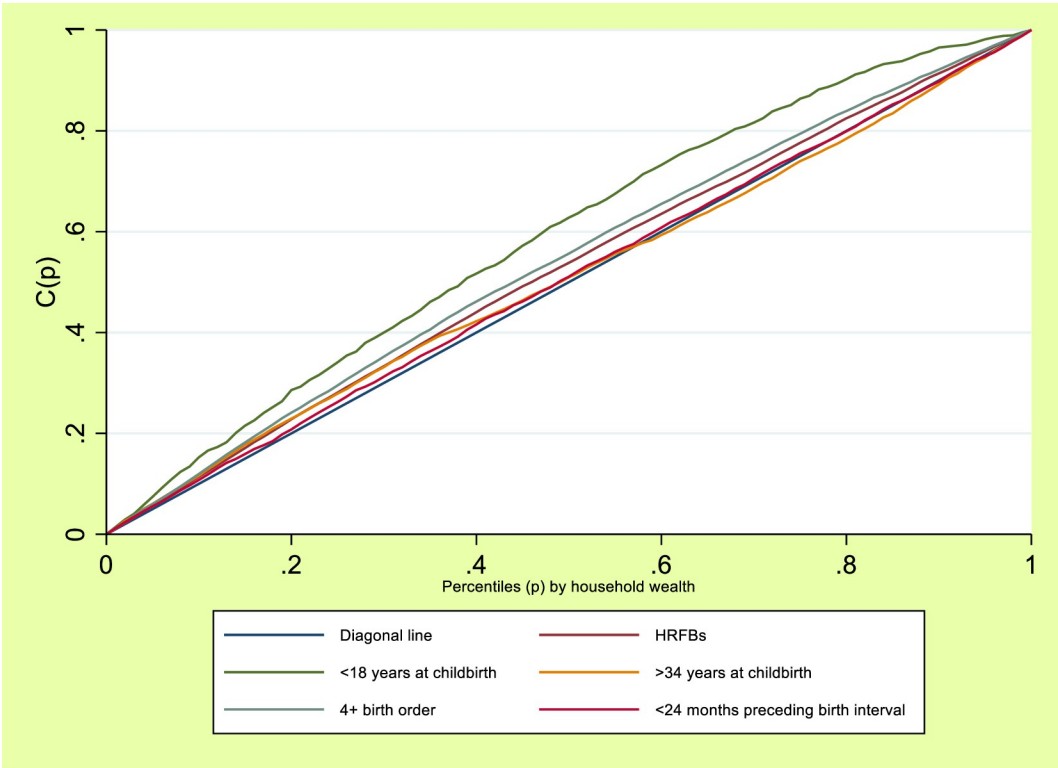

**Fig 1. Lorenz curve for HRFBs by household wealth gradient.**

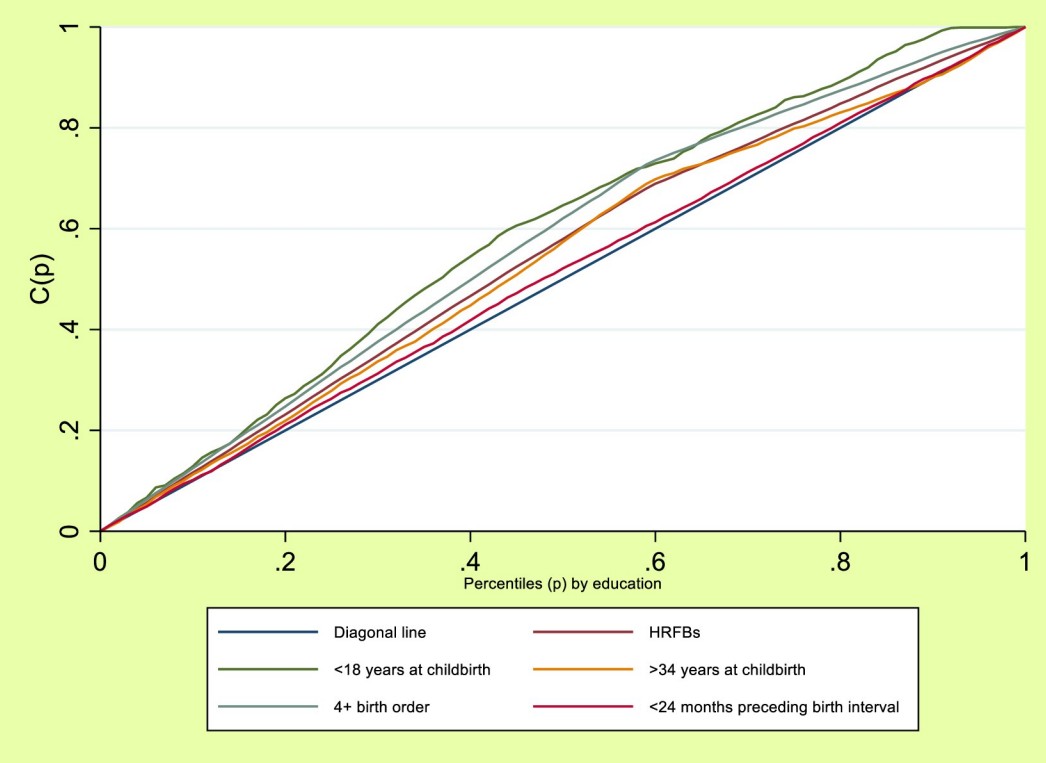

**Fig 2. Lorenz curve for HRFBs by education gradient.**

negative contributors to HRFBs among women of reproductive age in Nigeria. In concentration index decomposition, positive contributors show how much they contribute to the inequality. A positive value (percentage contribution) indicates that the variable is contributing to the observed inequality in HRFB. See Table 3 for the details.

## Discussion

This study has become the foremost to examine socioeconomic inequalities in HRFB among Nigerian women. The key findings showed high prevalence of HRFB with more concentration among the poor and uneducated women. The prevalence of HRFBs in Nigeria (63.5%) was consistent with the findings from Ethiopia [25, 42, 43], Kenya (70.86%) [44], Democratic Republic of the Congo (68.5%) [45], and African region [31], but higher than the pooled estimate from SSA countries (22.53%) [32] and East Africa region (57.6%) [46] respectively. Women from low-income households often lack access to quality healthcare and family planning services, leading to higher fertility rates and shorter birth intervals. Wealth provides women with financial independence and bargaining power within households, which can lead to better reproductive health outcomes. Conversely, poor women or those engaged in low-paying jobs are more vulnerable to HRFB due to limited resources and autonomy.

To develop effective health programmes to mitigate HRFB requires an understanding of the factors that predict the observed socioeconomic inequality [47]. This research sought to quantify and identify variables linked to socioeconomic inequality in HRFBs among Nigerian women, using the concentration index approach, given that the country's HRFBs foreshadow danger to the achievement of SDG 5.3 by 2030. Women's sexual and reproductive health is a global issue that is covered by several international programmes and policies, such as the

**Table 2. Household wealth and educational inequalities in HRFBs among Nigerian women.**

| Variable | Household wealth | | Education | |
|---|---|---|---|---|
| | Concentration Index (SE) | P | Concentration Index (SE) | P |
| **Family motility** | | 0.497 | | 0.647 |
| < 5 years | -0.038 (0.011)* | | -0.083 (0.010)* | |
| 5+ years (native) | -0.044 (0.003)* | | -0.079 (0.003)* | |
| **Age (years)** | | <0.001 | | <0.001 |
| 15–19 | -0.047 (0.010)* | | -0.040 (0.010)* | |
| 20–24 | -0.059 (0.014)* | | -0.099 (0.014)* | |
| 25–29 | -0.106 (0.007)* | | -0.158 (0.007)* | |
| 30–34 | -0.090 (0.005)* | | -0.143 (0.005)* | |
| 35–39 | -0.028 (0.003)* | | -0.046 (0.003)* | |
| 40–44 | -0.000 (0.000) | | 0.000 (0.000) | |
| 45–49 | 0.000 (0.000) | | 0.000 (0.000) | |
| **Region** | | 0.181 | | <0.001 |
| North Central | -0.033 (0.008)* | | -0.104 (0.007)* | |
| North East | -0.029 (0.006)* | | -0.056 (0.005)* | |
| North West | -0.028 (0.004)* | | -0.042 (0.003)* | |
| South East | -0.0183 (0.009)* | | -0.050 (0.007)* | |
| South South | -0.011 (0.010) | | -0.098 (0.009)* | |
| South West | -0.041 (0.012)* | | -0.127 (0.011)* | |
| **Residence** | | <0.001 | | <0.001 |
| Urban | -0.073 (0.005)* | | -0.111 (0.005)* | |
| Rural | -0.043 (0.003)* | | -0.078 (0.003)* | |
| **Religion** | | 0.082 | | <0.001 |
| Christian | -0.028 (0.005)* | | -0.087 (0.005)* | |
| Islam | -0.039 (0.003)* | | -0.061 (0.003)* | |
| Traditional/others | -0.073 (0.030)* | | -0.106 (0.028)* | |
| **Marital status** | | <0.001 | | 0.005 |
| Single | 0.019 (0.035) | | -0.143 (0.028)* | |
| Married | -0.051 (0.003)* | | -0.089 (0.003)* | |
| **Ever used contraceptive method** | | 0.442 | | 0.378 |
| No | -0.054 (0.003)* | | -0.097 (0.003)* | |
| Yes | -0.049 (0.005)* | | -0.102 (0.005)* | |
| **Covered by health insurance** | | 0.036 | | 0.983 |
| No | -0.050 (0.003)* | | -0.095 (0.003)* | |
| Yes | -0.093 (0.020)* | | -0.095 (0.020)* | |
| **Media use** | | <0.001 | | <0.001 |
| No | -0.021 (0.004)* | | -0.047 (0.003)* | |
| Yes | -0.050 (0.004)* | | -0.106 (0.004)* | |
| **Employment status** | | 0.033 | | 0.670 |
| No | -0.063 (0.005)* | | -0.098 (0.004)* | |
| Yes | -0.050 (0.003)* | | -0.101 (0.003)* | |
| **Sex of household head** | | 0.051 | | <0.001 |
| Male | -0.052 (0.003)* | | -0.089 (0.003)* | |
| Female | -0.034 (0.010)* | | -0.126 (0.010)* | |
| **Socioeconomic disadvantaged status** | | 0.123 | | <0.001 |
| Low | -0.033 (0.006)* | | -0.090 (0.006)* | |
| Medium | -0.032 (0.006)* | | -0.100 (0.005)* | |

(*Continued*)

**Table 2.** (Continued)

| Variable | Household wealth | | Education | |
|---|---|---|---|---|
| | Concentration Index (SE) | P | Concentration Index (SE) | P |
| High | -0.021 (0.004) | | -0.036 (0.003)* | |
| **Partner education** | | 0.002 | | <0.001 |
| No education | -0.012 (0.004)* | | -0.010 (0.002)* | |
| Primary | -0.012 (0.007) | | -0.016 (0.006)* | |
| Secondary | -0.008 (0.006) | | -0.073 (0.005)* | |
| Higher | -0.039 (0.009) | | -0.106 (0.009)* | |
| Do not know | -0.078 (0.009)* | | -0.072 (0.021)* | |
| **Total estimate** | -0.051 (0.003) | <0.001 | -0.096 (0.003) | <0.001 |

* Significant at p<0.05. p = comparing concentration indices across the levels of a variable. SE, standard error.

SDGs [48, 49]. Early pregnancy is one of the indicators of HRFB. Due to the strong cultural traditions and dire economic conditions in Nigeria, early marriage is encouraged in Nigeria for financial incentives directly and indirectly, as marriage frequently draws a dowry for the bride's family [50]. In many rural communities, parents even force young girls into marriage because they think it will protect their daughters from being sexually abused [51]. Among low socioeconomic populations, teenage pregnancy is primarily caused by early marriage. Reducing early pregnancies can be achieved by taking steps to delay marriage. Thus, in order to help achieve SDG 5.3 by 2030, family planning initiatives must include the goal of ending child marriage [50]. From a policy perspective, teens who are at risk of getting pregnant are the relevant group, which makes this crucial.

The results showed that women from low socioeconomic backgrounds had higher prevalence of HRFB than those from high socioeconomic status. HRFB concurs with this finding [52]. Findings further indicate the positive contributors to observed socioeconomic inequalities of HRFB. This suggests that poverty and lack of education makes women susceptible to HRFB [53]. Additionally, HRFB can be linked to the prevalent religious and cultural traditions

**Table 3. Decomposition of HRFBs among Nigerian women.**

| Variable | Elasticity | Concentration Index | Absolute Contribution | % Contribution |
|---|---|---|---|---|
| Family motility | 0.0442 | -0.0281 | -0.0050 | 3.7890 |
| Age (years) | 0.5407 | 0.0089 | 0.0192 | -14.6237 |
| Region | -0.0515 | 0.0711 | -0.0146 | 11.1724 |
| Residence | 0.0270 | -0.0007 | -0.0001 | 0.0601 |
| Education | -0.0596 | 0.2771 | -0.0660 | 50.3997 |
| Household wealth | -0.0335 | 0.2665 | -0.0357 | 27.2811 |
| Religion | 0.0916 | -0.0496 | -0.0182 | 13.8636 |
| Marital status | 0.0000 | -0.0010 | 0.0000 | 0.0000 |
| Ever used contraceptive method | 0.0137 | 0.2094 | 0.0115 | -8.7723 |
| Covered by health insurance | -0.0006 | 0.4735 | -0.0011 | 0.8678 |
| Media use | -0.0102 | 0.1449 | -0.0059 | 4.5064 |
| Employment status | 0.0007 | 0.0435 | 0.0001 | -0.0929 |
| Sex of household head | -0.0362 | 0.0019 | -0.0003 | 0.2108 |
| Socioeconomic disadvantaged status | 0.0491 | -0.0996 | -0.0195 | 14.9163 |
| Partner education | -0.0109 | 0.2138 | -0.0093 | 7.1351 |

that discourage women from engaging in premarital sex and modern contraceptive use [53]. More specifically, social stigma restricts access to reproductive health services [54], thereby leading to unplanned pregnancies, lack of optimal birth spacing or having large number of children. The problems associated with early pregnancies and suboptimal birth spacing among the disadvantaged should therefore be addressed through interventions that are considerate of geographical differences [52]. Healthcare stakeholders can create awareness by conducting community outreach programmes in addition to providing counselling on the risks associated with HRFB.

Empowerment through formal education is required to address the high incidence of HRFB among underprivileged women in Nigeria. Women who are empowered are more likely to plan their pregnancies or delay getting married [55]. Women with lower levels of education had a higher likelihood of experiencing HRFB than those with greater education [56]. This is likewise true that women who have higher educational attainment are protected from unintended pregnancies because of the empowerment that comes with getting a higher education [53]. Undoubtedly, education is important since people with less or no education are more likely to become pregnant at a younger age than people with a greater level of education [56]. It is necessary to implement a focused formal education intervention given the high prevalence of HRFB in certain geographic areas in Nigeria, which is caused by low levels of schooling and severe poverty. Improved socioeconomic status would assist in ending the generational cycle of poverty [55] and reduce the risk of early pregnancy, sub-optimal birth interval and high fertility rate [50].

## Strengths and limitations

This study utilized nationally representative data from the 2018 NDHS, making its findings of plausible comparison. It revealed socioeconomic disparities in HRFB and identified the contributory factors. These results provided policymakers with a chance to improve the delivery of counseling, health information and programmes to address the observed HRFB inequalities, by focusing on these identified factors. However, this study the decomposition analysis did not establish a cause-and-effect relationship between exploratory and outcome variables; rather, it predominantly signifies correlation. HRFB was evaluated based on self-reported data, potentially introducing recall or social desirability bias. Moreover, the assets-based wealth index, used as a proxy for household economic status, may not always provide accurate results compared to direct measurements of income and expenditure where such data are available or can be collected reliably. As HRFB could be due to women's bargaining power in the household as well as the society, however, the role of partners in HRFB was outside the scope of this study, as partners might also have a pivotal influence on HRFB.

## Conclusion

The study revealed high prevalence of HRFB and established socioeconomic inequalities in HRFB among Nigerian women. Addressing socioeconomic inequalities in HRFB requires comprehensive and multifaceted interventions. Investing in women's education and promoting wealth can significantly reduce HRFB. Education programs should aim to keep girls in school longer, delay the age of marriage, and increase their awareness of reproductive health. Enhancing economic opportunities for women through job creation, vocational training, and microfinance programs can provide women with financial independence and reduce HRFB. Economic empowerment enables women to make informed reproductive choices and access necessary healthcare services. Strengthening the healthcare system to provide accessible and affordable maternal healthcare services is crucial. This includes improving infrastructure in

rural areas, subsidizing healthcare costs for low-income women, and ensuring the availability of family planning services. Engaging community leaders and stakeholders in promoting reproductive health education and changing harmful cultural norms is essential. Community-based interventions can foster supportive environments for women's reproductive rights and health. Policymakers should prioritize reducing socioeconomic disparities in reproductive health. This involves implementing policies that address poverty, improve educational opportunities for girls, and ensure universal access to healthcare services.

## Acknowledgments

The authors appreciate the Demographic and Health Survey (DHS) for the approval and access to the original data.

## Author Contributions

**Conceptualization:** Michael Ekholuenetale.

**Data curation:** Michael Ekholuenetale.

**Formal analysis:** Michael Ekholuenetale.

**Investigation:** Michael Ekholuenetale.

**Methodology:** Michael Ekholuenetale.

**Project administration:** Michael Ekholuenetale.

**Resources:** Michael Ekholuenetale.

**Software:** Michael Ekholuenetale.

**Supervision:** Michael Ekholuenetale.

**Validation:** Michael Ekholuenetale.

**Visualization:** Michael Ekholuenetale.

**Writing – original draft:** Michael Ekholuenetale.

**Writing – review & editing:** Michael Ekholuenetale.

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
