## [Editor Report · Decision Letter 0]

16 Jun 2024

PONE-D-24-23186

Socioeconomic inequalities in high risk fertility behaviour among Nigerian women: a non-experimental population-based study

PLOS ONE

Dear Dr. Ekholuenetale,

Thank you for submitting your manuscript to PLOS ONE. After careful consideration, we have decided that your manuscript does not meet our criteria for publication and must therefore be rejected.

Specifically:

The article deals with studies of socioeconomic inequalities among certain communities. This type of study, methodology and results may be suitable for some other Journal.

I am sorry that we cannot be more positive on this occasion, but hope that you appreciate the reasons for this decision.

Kind regards,

Rajesh Sharma

Academic Editor

PLOS ONE

Additional Editor Comments :

The article deals with studies of socioeconomic inequalities among certain communities. This type of study, methodology and results may be suitable for some other Journal.

- - - - -

---

## [Author Response · Author response to Decision Letter 0]

20 Jun 2024

June 20, 2024

Manuscript ID: PONE-D-24-23186

Title: “Socioeconomic inequalities in high risk fertility behaviour among Nigerian women: a non-experimental population-based study”

RE: Response to Editor’s comments

Dear Editor, 

This letter is in reference to your email dated June 16, 2024 with editor’s comments. I am happy that my manuscript is being consider for review and possible publication. I am hoping for a positive outcome in the review process. 

Please do not hesitate to get in touch if you require any further information. 

Michael Ekholuenetale

Corresponding Author 

Editorial Comments:

The article deals with studies of socioeconomic inequalities among certain communities. This type of study, methodology and results may be suitable for some other Journal.

Response: I am happy that my appeal has been granted and the paper being considered for review and possible publication. Thank you very much.

---

## [Decision Letter · Decision Letter 1]

12 Sep 2024

PONE-D-24-23186R1Socioeconomic inequalities in high risk fertility behaviour among Nigerian women: a non-experimental population-based studyPLOS ONE

Dear Dr. Michael Ekholuenetale,

Thank you for submitting your manuscript to PLOS ONE. After careful consideration, we feel that it has merit but does not fully meet PLOS ONE’s publication criteria as it currently stands. Therefore, we invite you to submit a revised version of the manuscript that addresses the points raised during the review process.

**ACADEMIC EDITOR: Please insert comments here and delete this placeholder text when finished.** Be sure to:Indicate which changes you require for acceptance versus which changes you recommendAddress any conflicts between the reviews so that it's clear which advice the authors should followProvide specific feedback from your evaluation of the manuscriptPlease ensure that your decision is justified on PLOS ONE’s publication criteria and not, for example, on novelty or perceived impact.

We look forward to receiving your revised manuscript.

Kind regards,

Abiodun Adanikin, Ph.D

Academic Editor

PLOS ONE

2. Please note that your Data Availability Statement is currently missing the repository name. If your manuscript is accepted for publication, you will be asked to provide these details on a very short timeline. We therefore suggest that you provide this information now, though we will not hold up the peer review process if you are unable.

Additional Editor Comments (if provided):

Reviewers' comments:

Reviewer's Responses to Questions

**Comments to the Author**

1. If the authors have adequately addressed your comments raised in a previous round of review and you feel that this manuscript is now acceptable for publication, you may indicate that here to bypass the “Comments to the Author” section, enter your conflict of interest statement in the “Confidential to Editor” section, and submit your "Accept" recommendation.

Reviewer #1: (No Response)

Reviewer #2: All comments have been addressed

Reviewer #3: (No Response)

2. Is the manuscript technically sound, and do the data support the conclusions?

Reviewer #1: Yes

Reviewer #2: Yes

Reviewer #3: Yes

3. Has the statistical analysis been performed appropriately and rigorously? 

Reviewer #1: I Don't Know

Reviewer #2: Yes

Reviewer #3: Yes

4. Have the authors made all data underlying the findings in their manuscript fully available?

Reviewer #1: Yes

Reviewer #2: Yes

Reviewer #3: Yes

5. Is the manuscript presented in an intelligible fashion and written in standard English?

Reviewer #1: Yes

Reviewer #2: Yes

Reviewer #3: Yes

6. Review Comments to the Author

Reviewer #1: 1. There seems to be over-usage of the word "poor" throughout the manuscript, especially with respect to healthcare systems. To many, this might be condescending, albeit being true. Instead use 'low-resource/resource constraints or similar connotations.

2. Inconsistent use of verb tense throughout the manuscript. Since the data is cross-sectional, and has already been collected, and the analysis has already been done, use past tense preferably.

3. The Introduction, Results and Discussion, an din fact the whole study while statistically sound, does not explicitly acknowledge that High Risk Fertility Behaviour could be due women's bargaining power in the household as well as the society. While the author(s) briefly mention gender and patriarchial norms in Nigerian society, it will be helpful, if it is mentioned in the limitations, that the role of partners in HRFB was outside the scope of this study, as partners might also have a pivotal influence on HRFB.

4. The fluidity of writing could be improved, in the introduction and discussion, with incorporation of some theoretical basis to the risk factors selected for analysis.

Reviewer #2: The manuscript presents an important finding of social importance in Nigeria. The author has analysed the dataset properly and the manuscript is written in an intelligible fashion. Although a secondary data analysis, the statistical robustness of analysing and the issue highlighted in the article makes it suitable for publication.

Reviewer #3: The study is nicely conceptualized and contributes to reducing HRFB of Nigerian women. I request to author to clarify on following comment:

1. What advantages does decomposition analysis have over multivariate regression analysis in examining socioeconomic inequalities in this study?

7. PLOS authors have the option to publish the peer review history of their article (what does this mean?). If published, this will include your full peer review and any attached files.

Reviewer #1: No

Reviewer #2: **Yes: **Aftab Ahmad

Reviewer #3: No

---

## [Author Response · Author response to Decision Letter 1]

13 Sep 2024

September 12, 2024

Manuscript ID: PONE-D-24-23186R1

Title: “Socioeconomic inequalities in high-risk fertility behaviour among Nigerian women: a non-experimental population-based study”

RE: Revised manuscript submission and response to reviewers’ comments

Dear Editor, 

This letter is in reference to your email dated September 12, 2024 with reviewers’ comments. I am very pleased that the manuscript is potentially acceptable for publication in PLOS ONE once the revisions have been carried out. 

I would like to thank the reviewers for these insightful and helpful comments and for giving me the chance to revise our manuscript. I believe the revised manuscript has been significantly improved and the reviewers’ comments have been addressed adequately. I think in its current form it will make a valuable contribution to the literature on this increasingly important topic. 

Please find for your kind consideration the followings:

1) A section-by-section response to the comments and suggestions of the reviewers (below). 

2) The revised manuscript, provided as a marked-up copy and/or a clean copy. 

I hope that these changes meet with your favourable consideration. Please do not hesitate to get in touch if you require any further information. 

Michael Ekholuenetale

Corresponding Author 

Reviewers' comments:

Reviewer #1: 1. There seems to be over-usage of the word "poor" throughout the manuscript, especially with respect to healthcare systems. To many, this might be condescending, albeit being true. Instead use 'low-resource/resource constraints or similar connotations.

Response: Thank you very much for the insightful comment. This has been corrected throughout the manuscript.

2. Inconsistent use of verb tense throughout the manuscript. Since the data is cross-sectional, and has already been collected, and the analysis has already been done, use past tense preferably.

Response: The verb form has now been corrected throughout the manuscript. Thank you very much for the insightful comment.

3. The Introduction, Results and Discussion, and in fact the whole study while statistically sound, does not explicitly acknowledge that High Risk Fertility Behaviour could be due women's bargaining power in the household as well as the society. While the author(s) briefly mention gender and patriarchial norms in Nigerian society, it will be helpful, if it is mentioned in the limitations, that the role of partners in HRFB was outside the scope of this study, as partners might also have a pivotal influence on HRFB.

Response: As recommended, the limitations to the study has been revised to indicate that the role of partners in HRFB was outside the scope of this study. Thank you very much for the insightful comment.

4. The fluidity of writing could be improved, in the introduction and discussion, with incorporation of some theoretical basis to the risk factors selected for analysis.

Response: Thank you very much for the insightful comment. The theoretical basis for the selection of risk factors of HRFB has now been included in the methods section. 

Reviewer #2: The manuscript presents an important finding of social importance in Nigeria. The author has analysed the dataset properly and the manuscript is written in an intelligible fashion. Although a secondary data analysis, the statistical robustness of analysing and the issue highlighted in the article makes it suitable for publication.

Response: Thank you very much for the insightful comment.

Reviewer #3: The study is nicely conceptualized and contributes to reducing HRFB of Nigerian women. I request to author to clarify on following comment:

1. What advantages does decomposition analysis have over multivariate regression analysis in examining socioeconomic inequalities in this study?

Response: To clarify the advantages of decomposition analysis over multivariate regression analysis in examining socioeconomic inequalities as in this study;

Decomposition analysis and multivariate regression are both useful in examining socioeconomic inequalities, but decomposition analysis offers several distinct advantages in this study. For example, a) breakdown of inequality: decomposition analysis provides a clearer breakdown of the contribution of education and wealth to overall inequality. This allows for a more detailed understanding of how each variable contributes to socioeconomic inequalities, something multivariate regression cannot easily offer; b) quantifying contribution: decomposition analysis quantifies how much of the inequality in high-risk fertility behaviour (HRFB) is attributable to differences in the distribution of socioeconomic factors versus differences in their effects. This provides insight into whether inequalities are driven more by structural factors or by unequal treatment or outcomes; c) focus on inequality: while multivariate regression primarily examines relationships between variables and an outcome, decomposition analysis specifically focuses on understanding inequality. It is better suited for examining the distribution of an outcome across different groups; d) explaining gaps: decomposition analysis is particularly useful for explaining gaps between groups by isolating how much of the gap is due to differences in characteristics; e) policy-relevant insights: by identifying the specific drivers of inequality, decomposition analysis provides policymakers with clearer insights into which factors to target for interventions to reduce inequalities. Multivariate regression might indicate relationships, but it does not as clearly prioritize interventions based on the contributions of different variables.

In contrast, multivariate regression is more general in assessing associations and interactions among multiple variables but does not provide the same level of specificity in isolating the drivers of inequality. Thank you very much.

---

## [Editor Report · Decision Letter 2]

30 Sep 2024

Socioeconomic inequalities in high-risk fertility behaviour among Nigerian women: a non-experimental population-based study

PONE-D-24-23186R2

Dear Dr. Ekholuenetale,

We’re pleased to inform you that your manuscript has been judged scientifically suitable for publication and will be formally accepted for publication once it meets all outstanding technical requirements.

Kind regards,

Abiodun Adanikin, Ph.D

Academic Editor

PLOS ONE
---

## [Editor Report · Acceptance letter]

7 Oct 2024

PONE-D-24-23186R2 

PLOS ONE

Dear Dr. Ekholuenetale, 

I'm pleased to inform you that your manuscript has been deemed suitable for publication in PLOS ONE. Congratulations! Your manuscript is now being handed over to our production team.

Kind regards, 

on behalf of

Dr. Abiodun Adanikin 

Academic Editor

PLOS ONE